# Inducing conformational preference of the membrane protein transporter EmrE through conservative mutations

**Maureen Leninger[†], Ampon Sae Her[†], Nathaniel J Traaseth***

Department of Chemistry, New York University, New York, United States

**Abstract** Transporters from bacteria to humans contain inverted repeat domains thought to arise evolutionarily from the fusion of smaller membrane protein genes. Association between these domains forms the functional unit that enables transporters to adopt distinct conformations necessary for function. The small multidrug resistance (SMR) family provides an ideal system to explore the role of mutations in altering conformational preference since transporters from this family consist of antiparallel dimers that resemble the inverted repeats present in larger transporters. Here, we show using NMR spectroscopy how a single conservative mutation introduced into an SMR dimer is sufficient to change the resting conformation and function in bacteria. These results underscore the dynamic energy landscape for transporters and demonstrate how conservative mutations can influence structure and function.

DOI: https://doi.org/10.7554/eLife.48909.001

## Introduction

Membrane transport proteins catalyze the movement of ions and molecules across the membrane by binding substrates on one side of the bilayer and undergoing conformational changes (*Jardetzky, 1966*; *Zhang et al., 2016*). Structural and functional experiments have shown support for the alternating access model in which a transporter samples at least two different conformations that expose the substrate to the inside and outside environment of the cell membrane. The presence of inverted structural repeats within a single polypeptide chain is widespread in membrane protein transporters (*Shimizu et al., 2004*; *Forrest and biology, 2013*) and has been proposed as a mechanism to enable alternating access exchange (*Forrest et al., 2008*). Proteins with structural repeats are thought to arise evolutionarily from the fusion of smaller membrane protein genes, such as three or four transmembrane (TM) domain transporters that associate into oligomers to achieve the functional state (*Xu et al., 2014*; *Bay and Turner, 2009*). One class of these proteins is the small multidrug resistance (SMR) family (*Schuldiner, 2014*; *Schuldiner, 2009*; *Paulsen et al., 1996*) found in bacteria and archaea that contain four TM domains and assemble into antiparallel homodimers or heterodimers which resemble the inverted repeat structure in larger transporters. Homodimer transporters such as the multidrug efflux pump EmrE are able to insert into two opposing directions in the membrane (i.e. dual topology) whereas heterodimers are comprised of two genes that each insert into the membrane in a single orientation (i.e. single topology). The latter form the paired SMR (pSMR) subfamily and are thought to arise from gene duplication and evolution of dual topology SMR genes (*Bay and Turner, 2009*; *Rapp et al., 2007*; *Rapp et al., 2006*; *Kolbusz et al., 2010*). The similarity of the quaternary structure displayed by the SMR family with the pseudo two-fold symmetry seen in larger transporters suggests a role for gene duplication and fusion of smaller genes to give rise to transporters commonly found in higher level organisms (*Yan, 2013*; *Lolkema et al., 2008*). The widespread prevalence of inverted repeats and divergent evolution from

*For correspondence:
traaseth@nyu.edu

†These authors contributed equally to this work

Competing interests: The authors declare that no competing interests exist.

**eLife digest** Cells are bound by a thin membrane layer that protects the cell's interior from the outside environment. Within this layer are various transporter proteins that control which substances are allowed in and out of the cell. These transporters actively move substances across the membrane by loading cargo on one side of the layer, then changing their structure to release it on the other side.

Membrane transporters are typically made up of multiple repeating units. In more complex transporters, the genetic sequence for each of these structural units is fused together into a single gene that codes for the protein. It is thought that the repeated pattern evolved from smaller membrane protein genes that had duplicated and fused together. But, what are the evolutionary advantages of having more complex transporters being produced from a single, fused gene? To investigate this, Leninger, Sae Her, and Traaseth examined a simple transporter found in *Escherichia coli* bacteria, called EmrE, which contains two identical protein subunits that associate together to transport toxic molecules across the membrane.

Experiments revealed that changing a single amino acid (the building blocks that make up proteins) in one of the two subunits to make them minimally different from each other, dramatically modified the transporter's structure and function. The subtle amino acid change disrupted the balance of inward- and outward-facing proteins. This altered the transporter's ability to remove toxic chemicals from *E. coli* and reduced the bacteria's resistance to drugs.

The effects of a minor change to one of the identical halves of the EmrE transporter demonstrates how sensitive membrane transporters are to mutations. Furthermore, this observation could help explain why evolution favored more complex transporters comprised of fused genes in which single amino acid changes can greatly alter how the transporter operates.

DOI: https://doi.org/10.7554/eLife.48909.002

homo-oligomeric proteins suggests that asymmetry between functional subunits may have a fitness advantage, such as for evolving substrate specificity and direction of transport.

This work aims to establish a structure-function correlation for how single mutations introduced into one subunit of the EmrE dimer – *a minimal heterodimer* – induce a shift in the conformational equilibrium between inward-open and outward-open states. EmrE is a native *E. coli* transporter that couples drug efflux of cationic/aromatic compounds with the proton gradient (*Schuldiner, 2009*). It has recently been shown that the coupling stoichiometry can proceed in a 1:1 or 2:1 proton:drug ratio, where the protons are bound by the glutamic acid residues at position 14 in the dimer (*Robinson et al., 2017*). Since EmrE consists of an asymmetric and antiparallel homodimer, the relative energies between inward-open and outward-open states are identical for the wild-type protein. NMR spectroscopy has been a sensitive technique to probe this asymmetry by revealing a separate set of signals in the spectrum for monomers A and B within the dimer (*Cho et al., 2014*; *Gayen et al., 2013*; *Morrison et al., 2012*). Determining the effect of a single mutation in the dimer would provide insight into the energy landscape by mimicking a primitive evolutionary event such as those that may have led to the pSMR subfamily. Previously, we discovered that conservative mutations located within or close to loop 2 of EmrE modulated the overall rate of conformational exchange between inward-open and outward-open states (*Gayen et al., 2016*). Loop 2 is comprised of a short stretch of residues (approximately Tyr53, Ile54, and Pro55) and adjoins TM2 and TM3 of EmrE (*Figure 1A*). Loop 2 of monomer A is located on the open side of the transporter and does not make intermolecular contacts with monomer B in the tetraphenylphosphonium bound form, while loop 2 from monomer B is proximal to loop 3 of monomer A and near the bottom of the substrate cavity formed by TM1-3 (*Figure 1A*) (*Chen et al., 2007*; *Vermaas et al., 2018*; *Ovchinnikov et al., 2018*). The differential contacts in the structure and distinct chemical shifts for each monomer suggest a minimal heterodimer might influence the conformational equilibrium. Indeed, our preliminary experiments showed that a heterodimer formed between wild-type and a mutant from loop 2 (I54L) or a mutant from the N-terminal region of TM3 (I62L) could induce a change in the conformational equilibrium between inward-open and outward-open states when bound to the high affinity compound tetraphenylphosphonium (*Gayen et al., 2016*). While these

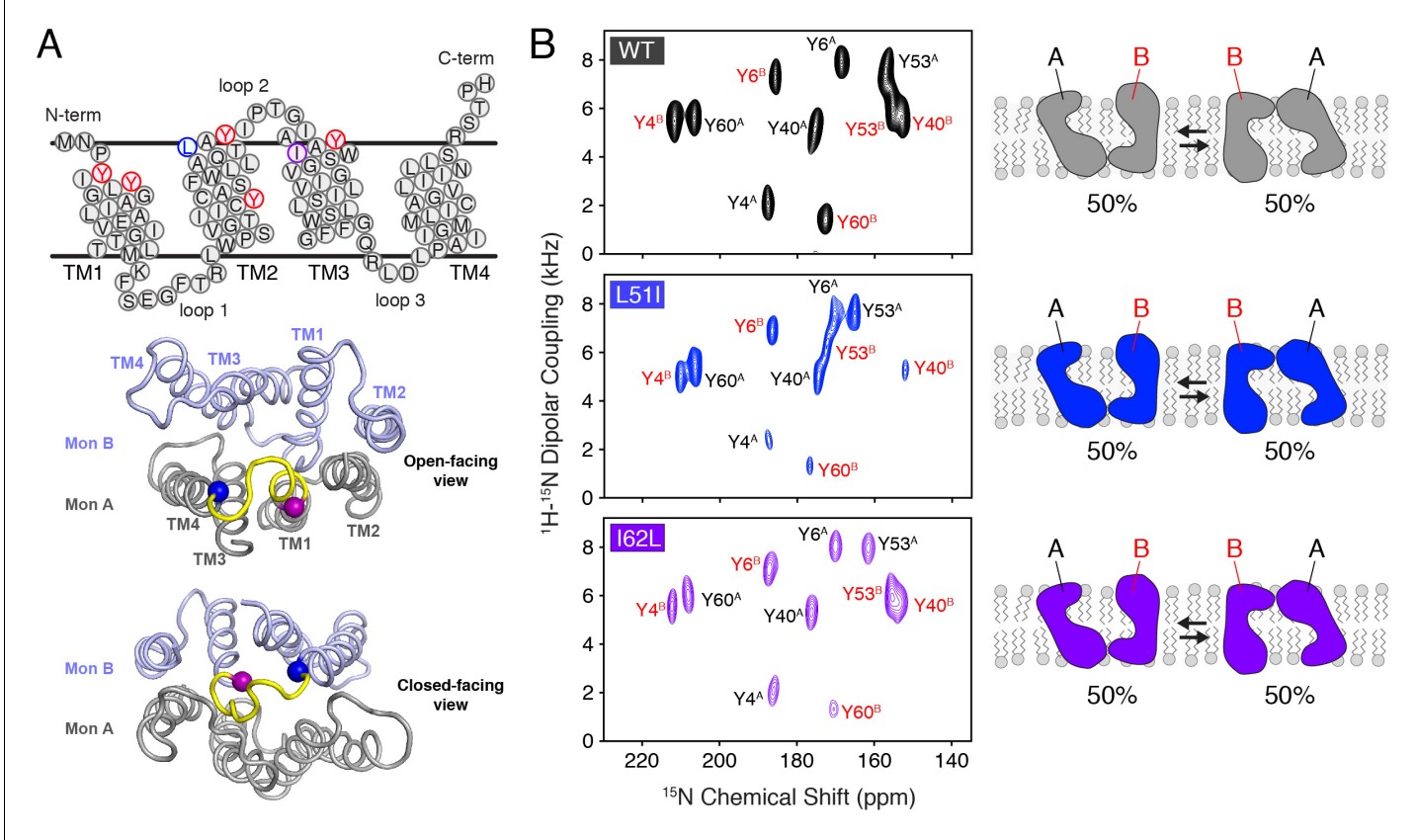

**Figure 1.** Conservative mutations within EmrE. (**A**) The top panel shows the primary sequence of EmrE and indicates the TM helices and loops. Residues colored in blue, purple, and red correspond to Leu51, Ile62, and tyrosine residues, respectively. The middle and bottom panels display a cartoon representation of the X-ray structure of EmrE (*3B5D*) (*Chen et al., 2007*) with the two monomers differentially colored: monomer A is in gray; monomer B is in light blue. Residues corresponding to residues 51–62 are highlighted in yellow and the blue and purple spheres correspond to Leu51 and Ile62, respectively. The middle panel shows a view of the open side of EmrE with residues 51–62 colored for monomer A. The bottom panel shows a view from the closed side of EmrE with the same residues colored for monomer B. (**B**) PISEMA spectra of wild-type EmrE, L51I, and I62L. All spectra are displayed at 5-times the standard deviation of the noise. Tyr53 located within loop 2 showed a notable spectral perturbation in L51I and I62L PISEMA spectra relative to the wild-type spectrum; a spectral expansion is displayed in *Figure 1—figure supplement 1*. Due to these perturbations, the assignment was confirmed using a single-site tyrosine mutation with a subsequent PISEMA dataset collected (see *Figure 1—figure supplement 2*).
DOI: https://doi.org/10.7554/eLife.48909.003

The following figure supplements are available for figure 1:

**Figure supplement 1.** Overlay of PISEMA spectra focusing on changes to Tyr53 upon mutation of loop 2 residues.
DOI: https://doi.org/10.7554/eLife.48909.004

**Figure supplement 2.** Assignment of Tyr53 using single-site mutagenesis.
DOI: https://doi.org/10.7554/eLife.48909.005

measurements support the presence of differential contacts of loop 2 within monomers A and B, no biological significance was offered for these findings or whether the conformational equilibrium was perturbed for other essential forms of the transporter needed to accomplish transport (i.e., proton-bound states, apo states).

Here, we report a conservative mutation (L51I) located in the C-terminal end of TM2 that has the greatest extent of altering the conformational equilibrium when paired with a wild-type monomer. The L51I/wild-type heterodimer was systematically investigated in different states of the transport cycle, including conditions where the essential anionic residue Glu14 was protonated or deprotonated and bound to drugs. These measurements revealed that both protonated and deprotonated drug-free forms of the L51I/wild-type heterodimer displayed a greater change in the conformational equilibrium relative to the drug-bound state. More importantly, growth inhibition experiments and efflux assays established that the conformational bias observed in vitro correlated with the functional

output in *E. coli*. These findings suggest a non-negligible role for conservative mutations in the duplication-divergence evolutionary theory (*Ohno, 1967*; *Taylor and Raes, 2004*) thought to govern the creation of larger membrane proteins from smaller genes.

## Results and discussion

### A single conservative mutant in the EmrE dimer alters the conformational equilibrium

Prior to investigating whether mutations in one subunit of an EmrE dimer could induce a change in the conformational equilibrium in vitro, NMR experiments were carried out on homodimer samples of wild-type EmrE and mutants (L51I, I62L). EmrE samples were prepared in magnetically aligned bicelles consisting of a 3.5/1 molar ratio of long chain (*O*-14:0-PC) and short chain (6:0-PC) phospholipids at a pH value of 5.8 that corresponds to the Glu14 proton-bound form of EmrE. Solid-state NMR experiments were carried out using the PISEMA experiment (*Wu et al., 1994*), since this technique gives the largest frequency separation between peaks corresponding to the two monomers within the asymmetric dimer (*Gayen et al., 2013*). PISEMA spectra of homodimer samples of wild-type, L51I, and I62L selectively labeled with $^{15}$N-tyrosine are shown in *Figure 1B*. Each spectrum displayed 10 peaks, which confirms the presence of twice the number of peaks as tyrosine residues in the primary sequence of EmrE (Tyr4, Tyr6, Tyr40, Tyr53, Tyr60), and is consistent with previous observations (*Gayen et al., 2013*; *Morrison et al., 2012*; *Gayen et al., 2016*; *Dutta et al., 2014b*).

To analyze whether heterodimers might lead to a preferred conformation, we prepared mixtures of wild-type EmrE with the L51I or I62L mutant. In each experiment, only the wild-type or mutant was isotopically enriched with $^{15}$N-tyrosine and mixed with its partner protein at natural abundance. Using this approach, only one protein was NMR active, while the other was NMR silent. PISEMA spectra corresponding to wild-type/L51I or wild-type/I62L where wild-type was isotopically enriched showed a primary set of intense peaks that corresponded to monomer B (*Figure 2A*). These signals were superimposable onto those peaks in the wild-type EmrE PISEMA spectrum (*Figure 2—figure supplement 1A*). Next, we carried out the reverse experiments where the mutants were isotopically enriched and wild-type was NMR silent. In these spectra, the isotopically enriched mutant (L51I or I62L) in the mixed dimer also showed an intense set of peaks (*Figure 2B*), yet these signals did not overlap with those of isotopically enriched wild-type in the mixed dimer (*Figure 2—figure supplement 1B*). The peak positions did however match onto the corresponding peaks in the L51I and I62L homodimer PISEMA spectra (*Figure 2—figure supplement 1A*). These data indicate that wild-type EmrE has a preference for monomer B in the heterodimer while L51I or I62L prefers monomer A for the proton-bound form of EmrE (*Figure 2C*). It is important to note that the mixture of wild-type EmrE and the L51I or I62L mutant produces a statistical fraction of heterodimers and homodimers in the samples. While our analysis focused on the most intense set of signals, we were able to resolve a weaker set of signals that likely corresponded to homodimers at a lower contour level (*Figure 2—figure supplement 2*). Nevertheless, these observations demonstrate that a minimal heterodimer comprised of a single conservative mutation can strongly influence the conformational equilibrium and perturb the energy landscape of EmrE.

### Assessing the conformational equilibrium for different states within the transport cycle

All experiments in *Figure 2* were performed on the proton-bound and drug-free form of EmrE. However, substrate transport requires EmrE to sample additional conformations within its catalytic cycle. To investigate how other states might impact the conformational equilibrium, we turned to solution NMR spectroscopy since it is a more sensitive technique to probe structure and conformational dynamics. To confirm the observations from PISEMA experiments in *Figure 2*, we collected solution NMR experiments of mixed dimers under proton-bound sample conditions in $^2$H-14:0-PC/$^2$H-6:0-PC (1/2 mol/mol) isotropic lipid bicelles. The mixed samples of wild-type EmrE and L51I were prepared in a similar manner as for solid-state NMR experiments with two notable differences: (1) the NMR active protein was isotopically enriched with $^{13}$C at the C$^\delta$ position of isoleucine residues and (2) the ratio of isotopically labeled wild-type EmrE to L51I at natural abundance was 1/3. The latter was possible due to the increased sensitivity of methyl detection in solution NMR spectroscopy. *Figure 3A*

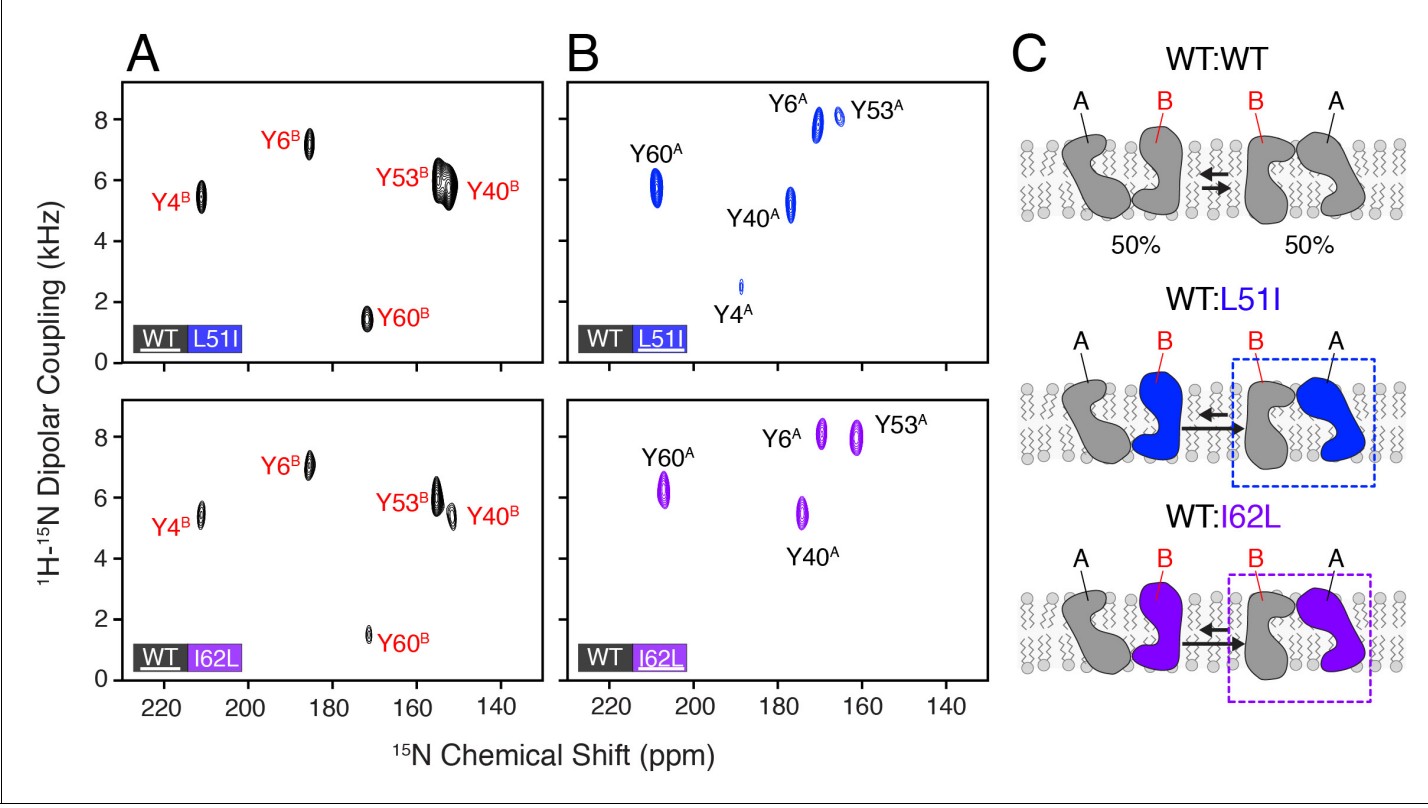

**Figure 2.** Heterodimer experiments showing conformational bias using oriented sample solid-state NMR. (**A**) PISEMA spectra of isotopically enriched wild-type EmrE mixed with L51I (top) or I62L (bottom) at natural abundance. (**B**) PISEMA spectra of isotopically enriched L51I (top) or I62L (bottom) mixed with wild-type EmrE at natural abundance. The underlined protein in each panel was isotopically enriched with $^{15}$N tyrosine, while the partner protein was unlabeled and NMR silent. Each PISEMA spectrum is shown at 5-fold above the standard deviation of the noise within the dataset. (**C**) Cartoon representation depicting how heterodimers lead to a change in the conformational equilibrium, where the mutant adopts monomer A conformation in the heterodimer and the wild-type monomer B population.

DOI: https://doi.org/10.7554/eLife.48909.006

The following figure supplements are available for figure 2:

**Figure supplement 1.** Overlay of PISEMA spectra of homodimers and heterodimers.
DOI: https://doi.org/10.7554/eLife.48909.007
**Figure supplement 2.** Evidence of homodimer peaks in PISEMA spectra of mixed dimer samples at a lower contour level.
DOI: https://doi.org/10.7554/eLife.48909.008

shows the $^{1}$H/$^{13}$C HMQC spectrum of isotopically enriched wild-type EmrE in a mixed dimer sample with L51I. From this spectrum, we observed an intense set of signals corresponding to monomer B, which confirms the results of our solid-state NMR experiments in aligned lipid bicelles. Thus, under sample conditions used to collect solution and solid-state NMR spectra (i.e. isotropic and aligned bicelles, respectively), the skewed conformational equilibrium is preserved.

Next, we sought to investigate whether other states of the transport cycle, such as deprotonation of Glu14 and drug bound forms, may alter the conformational equilibrium. Previously, we reported pH induced chemical shift perturbations within EmrE's NMR methyl spectrum that were centered at a pH value of 7.0 and corresponded to a biologically relevant $pK_a$ of Glu14 (*Gayen et al., 2016*). Thus, we acquired a $^{1}$H/$^{13}$C HMQC spectrum with a mixed sample of $^{13}$C enriched EmrE and natural abundance L51I at a pH value of 9.1. Similar to the proton-bound dataset, this spectrum showed intense peaks that corresponded to the signals stemming from monomer B (*Figure 3B*). This is in stark contrast to the spectrum of wild-type EmrE at high pH that showed approximately equal intensities of monomer A and B peaks (*Figure 3B*). This result supports the conclusion that the skewed conformational equilibrium is preserved under conditions in which Glu14 is deprotonated. Note that we also observed monomer A signals at a lower contour level, which arises due to the statistical

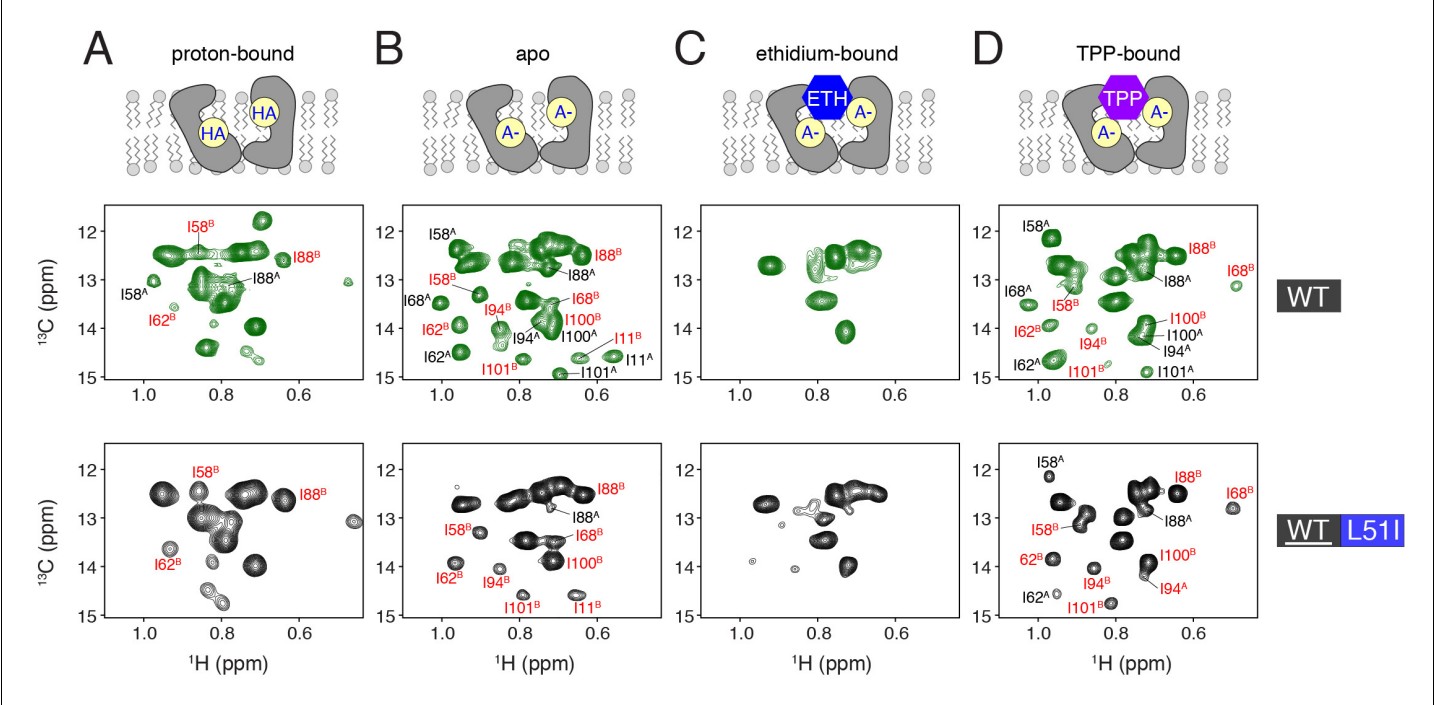

**Figure 3.** Determination of conformational equilibria among different states within the transport cycle probed with solution NMR. HMQC spectra of wild-type (middle row, green spectra) or isotopically enriched wild-type mixed with 3-fold excess L51I at natural abundance (bottom row, black spectra). The sample conditions were: (A) proton-bound (pH = 5.6), (B) deprotonated, apo (pH = 9.1), (C) ethidium-bound (pH = 9.1), and (D) tetraphenylphosphonium-bound (pH = 9.1). In each case, the isotopically enriched protein was $^{13}$C labeled at the C$^\delta$ position of isoleucine. Wild-type spectra serve as a reference since the population between monomers A and B is 50/50. Peaks labeled with red correspond to monomer B peaks, while those in black correspond to monomer A peaks.

DOI: https://doi.org/10.7554/eLife.48909.009

The following figure supplement is available for figure 3:

**Figure supplement 1.** Evidence of homodimer peaks in solution NMR mixed dimer samples at a lower contour level.

DOI: https://doi.org/10.7554/eLife.48909.010

nature of mixing as discussed above (*Figure 3—figure supplement 1A*). Using the relative peak intensities of A and B signals in the spectrum, we estimated the equilibrium population of wild-type EmrE to assume monomer B in the heterodimer at ~96% (range of 90% to 99% based on standard deviation). This value corresponds to a free energy of ~1.8 kcal/mol induced by the L51I mutation in the heterodimer.

To determine the effect of drug binding on the conformational equilibrium, we acquired NMR spectra for the mixed dimers in the presence of ethidium and tetraphenylphosphonium. These drug substrates are commonly used in the EmrE literature for measuring resistance, transport, and binding (*Yerushalmi et al., 1995*; *Curnow et al., 2004*; *Robinson et al., 2018*). Addition of ethidium induced a significant amount of spectral broadening and ablation of peak intensities in both the wild-type sample and that of the mixed dimer where wild-type was isotopically enriched (*Figure 3C*). These data suggest intermediate chemical exchange, which likely stems from motion corresponding to exchange between inward-open and outward-open states (*Cho et al., 2014*). From these data, it is likely that the equilibrium is less skewed in the mixed sample upon ethidium binding. Namely, if the populations were maintained to the same extent as in the mixed dimer sample at pH 9.1 (*Figure 3B*), the effect of intermediate chemical exchange would not induce peak broadening beyond detection. Hence, the effect of ethidium binding suggests that the conformational bias is reduced when the heterodimer is bound to a drug substrate. Furthermore, the line-broadening supports the presence of drug-induced dynamics in the heterodimer, which would enable conformational exchange needed for drug transport.

Finally, we investigated the effect of tetraphenylphosphonium binding on the conformational equilibrium. Unlike ethidium, tetraphenylphosphonium binds with greater affinity to EmrE and significantly reduces the rate of conformation exchange (i.e. slow chemical exchange regime) (*Cho et al., 2014*; *Morrison and Henzler-Wildman, 2014*). We carried out the same NMR experiment by adding tetraphenylphosphonium to a mixed dimer sample of $^{13}C^{\delta}$-Ile labeled EmrE mixed with natural abundance L51I (*Figure 3D*). Unlike the effect of ethidium binding, the spectra were nicely resolved in the presence of tetraphenylphosphonium. We observed that monomer B peaks were more intense relative to those of monomer A; however, the signal intensities stemming from monomer A were significantly greater compared to any of the drug-free samples (see I58[A], I62[A], and I88[A] peaks in *Figure 3D*). Specifically, the ratio of monomer B to A peak intensities were reduced upon addition of tetraphenylphosphonium relative to the deprotonated sample. Using the relative peak intensities of A and B, we estimated the equilibrium population of wild-type to assume monomer B in the heterodimer at ~86% (standard deviation range of 79% to 92%), which corresponds to ~1.1 kcal/mol induced by the single mutant in the heterodimer. This result is in agreement with the ethidium binding experiment. These data also provide evidence that drug binding would not *trap* the transporter into a single conformation in the transport cycle and is consistent with previous work proposing the substrate controls the rate of conformational exchange (*Morrison and Henzler-Wildman, 2014*).

## Functional assays in *E. coli* reveal biological significance for equilibrium changes

*E. coli* growth inhibition assays against ethidium bromide were carried out to determine the biological significance of our NMR observations. Initially, we tested whether the mutations L51I and I62L were able to confer resistance when expressed individually. *Figure 4A* shows a growth inhibition assay using serial 10-fold dilutions on an LB agar plate with wild-type, L51I, I62L, and a control vector. From these data, L51I and I62L displayed a strong phenotype toward ethidium and were indistinguishable relative to wild-type EmrE. Thus, each mutation appears to be fully functional and able to couple to the electrochemical potential to accomplish active drug efflux.

To probe the effect of heterodimer induced changes to the conformational equilibrium, we designed transporter complementation experiments by expressing two genes: (1) gene 1 consisted of wild-type EmrE, L51I, or I62L and (2) gene 2 consisted of single topology variants of EmrE. The latter utilized similar constructs as previous work showing insertion topology can be controlled by mutating arginine and lysine residues within the primary sequence (*Rapp et al., 2007*) (i.e. the positive inside rule; *Heijne, 1986*). Using this technology, gene 2 consisted of N- and C-termini facing the cytoplasmic direction (EmrE[in]) or periplasmic direction (EmrE[out]), respectively. Resistance assays verified literature results (*Rapp et al., 2007*) that EmrE[in] and EmrE[out] expressed individually were unable to confer a phenotype to ethidium, whereas co-expression displayed a strong phenotype toward this compound (*Figure 4B*).

Next, we carried out resistance assays using mixtures of wild-type, L51I, or I62L with EmrE[in] or EmrE[out]. The experiments where wild-type EmrE was co-expressed with EmrE[in] or EmrE[out] revealed identical levels of conferred resistance relative to wild-type EmrE alone (*Figure 5A*). To ensure that wild-type was forming a heterodimer with EmrE[in] or EmrE[out], we designed a control experiment in which an additional mutation (E14Q) was engineered into the EmrE[in] or EmrE[out] constructs (i.e. E14Q[in] or E14Q[out]). E14Q was selected since EmrE requires two glutamic acid residues to confer drug resistance (*Rapp et al., 2007*). Thus, if a dimer of wild-type and E14Q[in] or E14Q[out] formed, we would observe a reduced growth phenotype. Indeed, the results of these experiments displayed growth inhibition when wild-type was co-expressed with E14Q[in] or E14Q[out] (*Figure 5A*). Taken together with the robust phenotype of wild-type co-expressed with EmrE[in] or EmrE[out], these results confirmed that the transporter complementation approach leads to heterodimer formation in the cell membrane.

We next carried out experiments with L51I and I62L co-expressed with EmrE[in] or EmrE[out]. Remarkably, when EmrE[in] was co-expressed with L51I or I62L, we observed a significant reduction in ethidium resistance on LB agar plates (*Figure 5B*). To the contrary, EmrE[out] co-expressed with L51I or I62L retained the same phenotype as the single-site mutant alone. To provide additional support, we carried out resistance assays in liquid media. These experiments were performed by growing *E. coli* cultures containing plasmids to an optical density at 600 nm of 1.0, then adding ethidium and recording the culture density as a function of time. Similar to the LB agar plate results, we found that

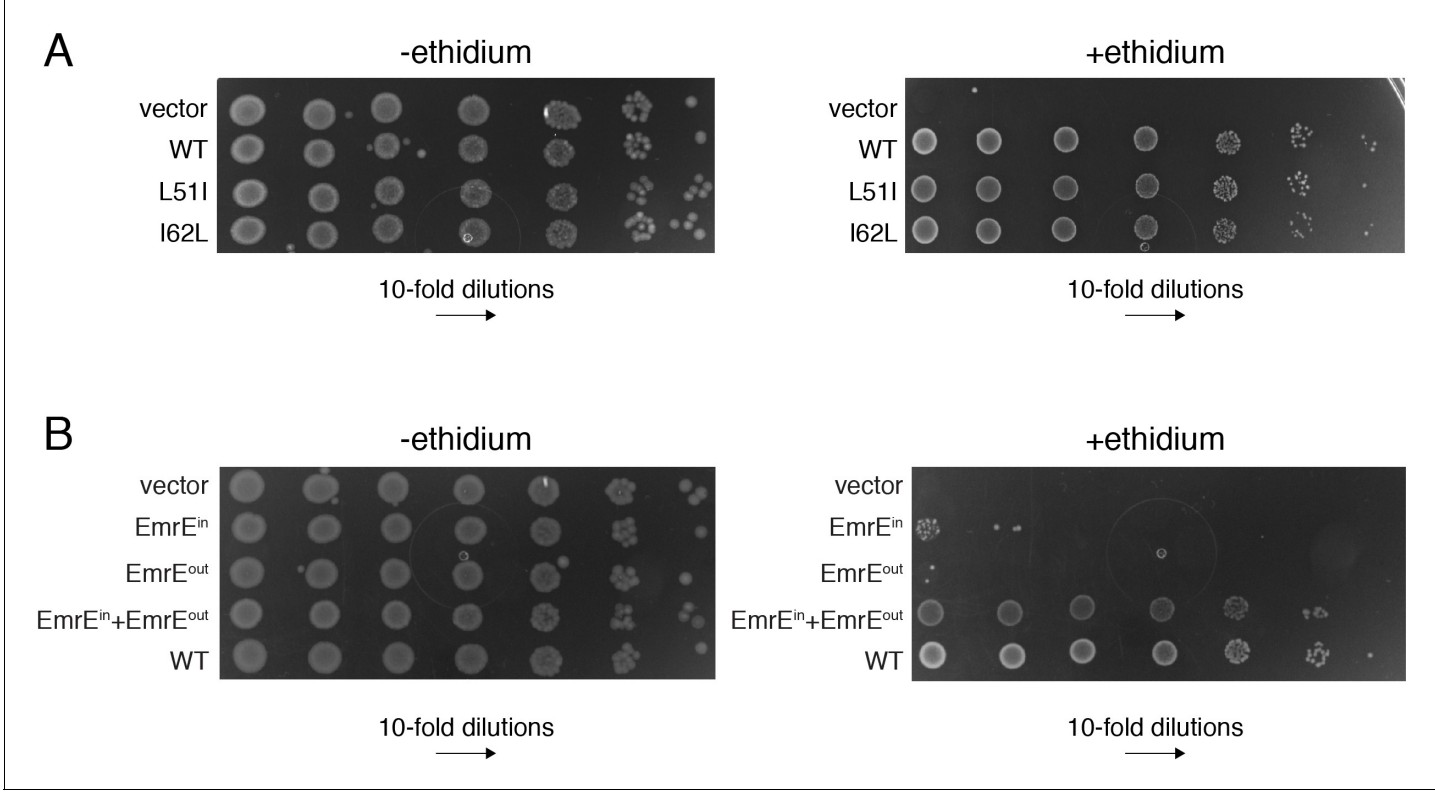

**Figure 4.** *E. coli* growth inhibition assays containing plasmids of wild-type EmrE, conservative mutants, and single topology variants of EmrE. (**A**) LB agar plates in the absence or presence of ethidium bromide spotted with *E. coli* containing plasmids corresponding to vector (control), wild-type EmrE, L51I, and I62L. Serial 10-fold dilutions are displayed in each panel from left to right. (**B**) LB agar plates in the absence or presence of ethidium bromide spotted with *E. coli* containing plasmids corresponding to vector (control), EmrE$^{in}$, EmrE$^{out}$, EmrE$^{in}$ and EmrE$^{out}$ co-expressed, and wild-type EmrE. Serial 10-fold dilutions are displayed in each panel from left to right. A schematic of the constructs and sequences of EmrE$^{in}$ and EmrE$^{out}$ are given in *Figure 4—figure supplement 1*.

DOI: https://doi.org/10.7554/eLife.48909.011

The following figure supplement is available for figure 4:

**Figure supplement 1.** Constructs used for resistance assays.

DOI: https://doi.org/10.7554/eLife.48909.012

the L51I or I62L mutant expressed with EmrE$^{in}$ displayed a faster drop in the optical density, which signified a reduced ability to confer resistance relative to the same mutant expressed with EmrE$^{out}$ (*Figure 5C*). It is important to note that if heterodimers of L51I or I62L and EmrE$^{in}$ did not form, we would have observed the same resistance phenotype of L51I or I62L alone. However, the reduced phenotype and the control experiments discussed above (*Figure 5A*; *Figure 5—figure supplement 1*) indicate a specific association in the transporter complementation experiments that are influenced by the conformational equilibrium within the assembled heterodimer in the cell membrane.

To provide direct evidence that the overall rate of ethidium transport was different between the mutant co-expressed with EmrE$^{in}$ and EmrE$^{out}$, we performed an ethidium efflux assay by measuring the intrinsic fluorescence of ethidium. *E. coli* were treated with ethidium bromide and the ionophore carbonyl cyanide *m*-chlorophenylhydrazone (CCCP), which causes the cytoplasmic ethidium concentration and fluorescence to increase. Upon addition of glucose and removal of CCCP, the membrane potential is reestablished, and the fluorescence decreases as ethidium is transported out of the cytoplasm. This assay was performed with L51I co-expressed with EmrE$^{in}$ or EmrE$^{out}$. Immediately following the addition of glucose, the fluorescence dropped ~3 fold faster for L51I co-expressed with EmrE$^{out}$ than L51I co-expressed with EmrE$^{in}$ (*Figure 5D*). Furthermore, the fluorescence value at steady state (3600 s) was significantly lower for L51I co-expressed with EmrE$^{out}$ compared to L51I co-expressed with EmrE$^{in}$ (*Figure 5E*). This indicates a reduced cytoplasmic ethidium concentration for the L51I/EmrE$^{out}$ sample. Taken together, these data are consistent with the resistance assay

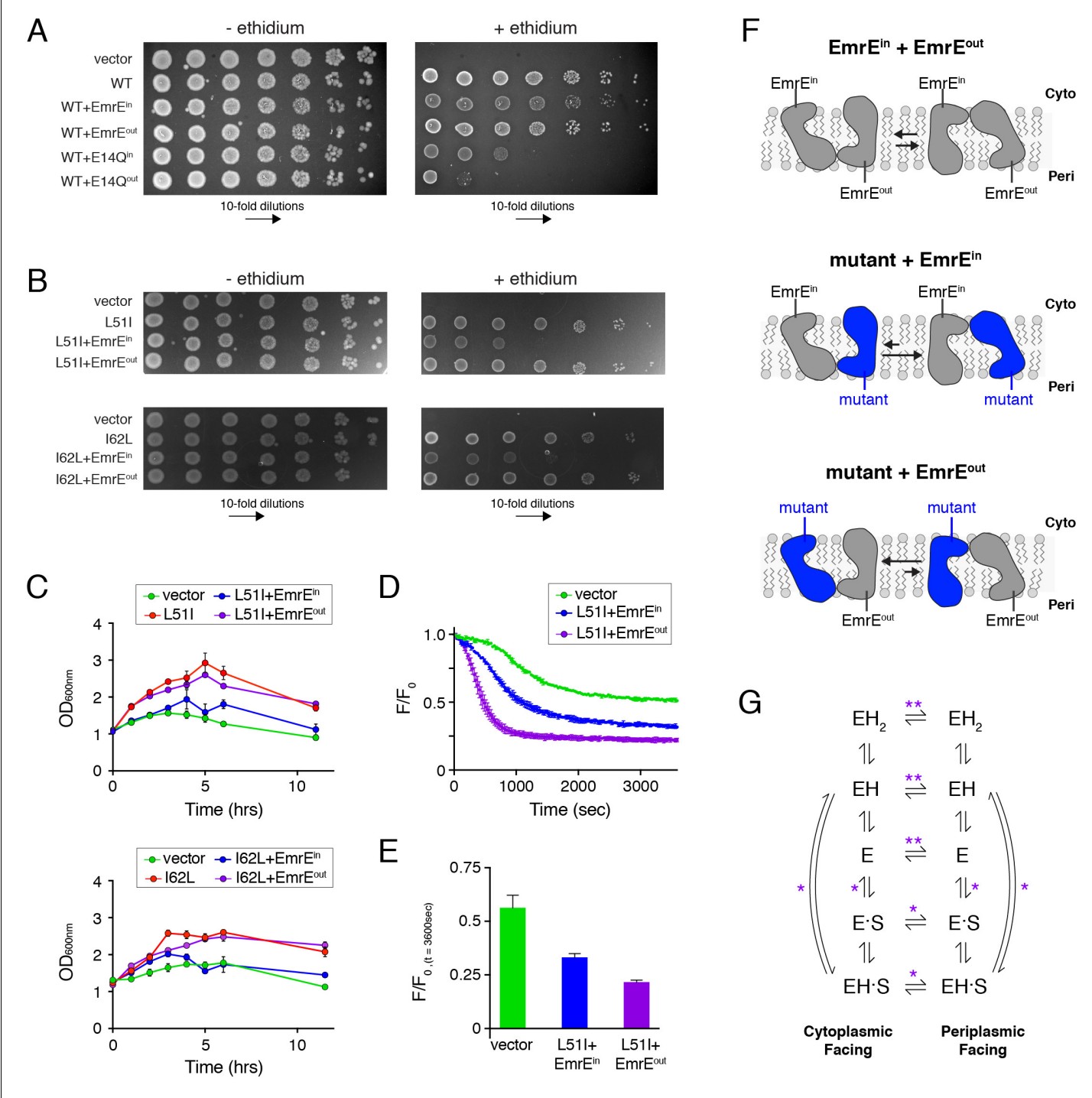

**Figure 5.** Assays to determine the biological significance of conformational bias observed in NMR experiments. (**A**) LB agar plates in the absence or presence of ethidium bromide spotted with *E. coli* containing plasmids corresponding to vector (control), wild-type EmrE, and combinations of wild-type EmrE co-expressed with EmrE[in], EmrE[out], E14Q[in], or E14Q[in]. Serial 10-fold dilutions are displayed in each panel from left to right. (**B**) LB agar plates in the absence or presence of ethidium bromide spotted with *E. coli* containing plasmids corresponding to vector (control), L51I, I62L, and combinations of L51I or I62L co-expressed with EmrE[in] or EmrE[out]. Serial 10-fold dilutions are displayed in each panel from left to right. (**C**) Resistance assays in liquid media in the presence of ethidium bromide (240 µg/mL). *E. coli* cultures containing the indicated plasmid were grown to an $OD_{600}$ of ~1.0 and treated with ethidium bromide. The point of ethidium addition represents time = 0. The error bars reflect the standard deviation from three trials. Error bars not observed are smaller than the points. (**D**) Ethidium efflux assays with *E. coli* transformed with plasmids corresponding to vector (control) and L51I co-expressed with EmrE[in] or EmrE[out]. The normalized ethidium fluorescence is plotted as a function of time following the addition of

*Figure 5 continued on next page*

*Figure 5 continued*

glucose at time zero. (**E**) The normalized fluorescence value from panel D at 3600 s. (**F**) Schematic depiction skewed conformational equilibria for the different heterodimers as indicated within the panel. (**G**) EmrE transport model proposed by *Robinson et al. (2017)*. 'E' refers to the EmrE dimer, 'H' is proton, and 'S' is the drug substrate. The states on the left are cytoplasmic-facing, while those on the right are periplasmic-facing. The double asterisks indicate equilibria most influenced by the heterodimers displaying conformational bias (drug-free states), while a single asterisk indicates equilibria influenced to a minor extent.

DOI: https://doi.org/10.7554/eLife.48909.013

The following figure supplements are available for figure 5:

**Figure supplement 1.** Resistance assays to test the association of EmrE$^{in}$ and EmrE$^{out}$ with L51I.

DOI: https://doi.org/10.7554/eLife.48909.014

**Figure supplement 2.** Numerical simulation of drug uptake into vesicles by heterodimers of EmrE and L51I.

DOI: https://doi.org/10.7554/eLife.48909.015

results and support the conclusion that a change in the conformational equilibrium can influence the overall rate of drug efflux. Note that a previous study reported an ethidium efflux rate difference of 1.6-fold between wild-type *E. coli* and an AcrAB-TolC efflux pump knockout strain that led to a 30-fold difference in the minimum inhibitory concentration (*Paixão et al., 2009*). This shows that somewhat modest transport rate differences can have deleterious effects on bacterial growth, which stems from increased ethidium accumulation that leads to irreversible binding to DNA (*Jernaes and Steen, 1994*; *Lambert and Le Pecq, 1984*).

## Rationalization for why a change in the conformational equilibrium influences phenotype

*How does the NMR observed conformational change of the heterodimer correlate with the functional results?* The transporter complementation experiments create topologically defined heterodimers due to the single topology of EmrE$^{in}$ and EmrE$^{out}$ and the antiparallel association of the dimer quaternary structure. Using this information and knowledge from NMR experiments that the mutant had a preference for monomer A in the heterodimer, we can make inferences about how a shift in the conformational equilibrium influences function. Specifically, the mutant paired with EmrE$^{out}$ in a heterodimer would favor the inward-open/cytoplasmic-facing conformation (*Figure 5F*). Likewise, the mutant paired with EmrE$^{in}$ will bias toward the outward-open/periplasmic-facing conformation under the same conditions (*Figure 5F*). Hence, the resistance assay results suggest that the conformational equilibrium favoring the outward-open/periplasmic state has a deleterious impact on the overall transport cycle. In contrast, a heterodimer with a conformation favoring the inward-open/cytoplasmic facing conformation gives no measurable reduction in phenotype. This finding is in harmony with our prior observation that the role of the pH gradient is to favor the inward-open conformation that is poised for drug binding (*Gayen et al., 2016*). Thus, reducing the effect of the pH gradient by biasing the equilibrium in the opposite direction apparently is sufficient to alter the net transport as observed in growth inhibition experiments and efflux assays.

It is important to underscore that the equilibrium constant between inward-open and outward-open states of the heterodimer ultimately stems from differential kinetic rate constants pertaining to the inward-open to outward-open and outward-open to inward-open transitions (see kinetic model [*Robinson et al., 2017*] in *Figure 5G*). Thus, we hypothesize that the primary kinetic steps in the transport cycle that influence the observed phenotype are the inward-open to outward-open and outward-open to inward-open rates for the drug-free heterodimer (*Figure 5G*, see double asterisks). We make this conclusion based on two observations. First, the populations were more skewed for the drug-free states than those for the drug bound states. Second, NMR experiments with ethidium indicate that the conformational exchange was not halted in the heterodimer. The latter means that the conformational bias does not result in a *locked-state* that would ultimately lead to loss-of-function (i.e. no substrate turnover).

To provide support for this conclusion, we performed a numerical simulation of pH-driven transport into liposomes using the kinetic model and parameters introduced by *Robinson et al. (2017)*. The differences were to change the inward-open to outward-open and outward-open to inward-open conformational exchange rates based on the populations observed in our heterodimer experiments and to account for the fact that drug binding affinities to wild-type(A)/mutant(B) and mutant

(A)/wild-type(B) are not necessarily the same (see *Figure 5—figure supplement 2*). Using these simulations, we found that the rate of drug transport was ~5.6 fold faster for a heterodimer of mutant and EmrE^out versus a heterodimer of mutant and EmrE^in (*Figure 5—figure supplement 2*). In addition, the former achieved a small but significant increase in the final concentration of drug within the vesicles, which further suggests that differences in kinetic rates in EmrE's transport cycle give rise to a change in the cytoplasmic drug concentration between heterodimers with opposite insertion topologies in *E. coli*. These calculations correlate with the enhanced phenotype and increased rate of ethidium efflux and provide evidence that shifting conformational equilibria via specific rate constants in a transport cycle has a direct impact on functional output.

## Conclusion

Despite the prominence of inverted repeat domains within transporters, there are only a few examples of mutational studies to influence conformational equilibria. These experiments involve the creation of loss-of-function, non-conservative mutations (e.g. Trp to Gly) at key structural contacts with the goal of crystallographically trapping particular conformations in the transport cycle (*Smirnova et al., 2013*; *Latorraca et al., 2017*). Our observations demonstrate that a minimal heterodimer comprised of a single conservative mutation is sufficient to disrupt the preferred resting conformation and underscore the presence of a dynamic energy landscape where relatively small energy differences exist among conformations in the transport cycle. From an evolutionary perspective, we hypothesize there might be fitness advantages for having separate genes that form a hetero-oligomer or a single polypeptide chain where the structural repeats are contained within one protein. In these cases, a single mutation can be introduced within the functional assembly that is not possible for a homo-oligomeric protein. Functional differences between closely related proteins stemming from one mutation in a hetero-oligomer versus multiple mutations in a homo-oligomer could potentially include influencing the propensity of a transporter to carry out antiport or symport. In fact, in addition to EmrE's known antiport activity, it is also able to carry out symport under certain conditions (*Robinson et al., 2017*), including import of polyamines upon mutation (*Brill et al., 2012*). Therefore, the observation that conservative mutations can display skewed conformational equilibria and influence function opens the possibility of somewhat modest evolutionary paths for achieving symport or antiport.

## Materials and methods

### Protein expression and purification

Protein expression and purification was carried out as previously reported (*Gayen et al., 2013*). EmrE is expressed as a fusion construct with maltose-binding protein in the pMAL vector (New England Biolabs Inc) in *E. coli* BL21(DE3) cells. For oriented sample NMR experiments, $^{15}$N-tyrosine labeled EmrE was expressed with IPTG for 4 hr at 25°C in an amino acid mixture consisting of 19 amino acids at natural abundance (300 mg/L) and $^{15}$N-tyrosine (120 mg/L). Unlabeled EmrE used in the heterodimer experiments was expressed in LB medium at natural abundance. For solution NMR experiments, wild-type and L51I were expressed in a fully perdeuterated background as previously described (*Gayen et al., 2016*). Wild-type protein was isotopically enriched with $^{13}$C at the $C^{\delta}$ position of isoleucine methyl groups with the addition of 50 mg/L 2-ketobutyric acid-4-$^{13}$C, 3,3-$^{2}$H$_2$ sodium salt hydrate 1 hr before induction. The bacterial cells following expression were lysed and the fusion protein was purified with amylose affinity chromatography. The fusion protein was cleaved with tobacco etch virus protease (TEV) and EmrE was further purified by size exclusion chromatography using a Superdex 200 10/300 column (GE Healthcare) in 0.06% w/v *n*-dodecyl-β-D-maltopyranoside (DDM).

### NMR sample preparation

For oriented sample solid-state NMR studies, purified EmrE in DDM detergent was reconstituted into 1,2-di-*O*-tetradecyl-sn-glycero-3-phosphocholine/dihexanoyl-sn-glycero-3-phosphocholine (*O*-14:0-PC/6:0-PC) bicelles at a molar ratio of 3.5/1 (i.e. q = 3.5). The bicelles were made with a protein concentration ~2 mM with a combined lipid concentration of 25% (w/v) in 80 mM HEPES and 20 mM NaCl at pH = 5.4. The heterodimer samples were prepared by mixing the two proteins (wild-type/

L51I or wild-type/I62L) with a molar ratio 1/1.2, where the protein in excess was at natural abundance (i.e. NMR silent). The two proteins used to prepare the heterodimer sample were incubated together at 37°C in the presence of 50 mM DTT for 1 hr immediately prior to reconstitution into lipids. The reconstitution into the bicelles was carried out in the same manner as those for homodimer samples (*Leninger and Traaseth, 2018*).

For solution NMR studies, purified EmrE in DDM detergent was reconstituted into dimyristoyl-sn-glycero-3-phosphocholine (14:0-PC) and dihexanoyl-sn-glycero-3-phosphocholine (6:0-PC) bicelles with a molar ratio of 1/2. The acyl chains of 14:0-PC and 6:0-PC were perdeuterated (Avanti Polar Lipids) to reduce the lipid signals in $^1H/^{13}C$ heteronuclear correlation experiments. The heterodimer samples were prepared by mixing isotopically enriched wild-type with natural abundance L51I in a 1/3 molar ratio. The two proteins used to prepare the heterodimer sample were incubated together at 37°C for 1 hr immediately prior to reconstitution into lipids. The heterodimers contained 0.533 mM total protein (0.133 mM wild-type, 0.4 mM L51I). The long-chain lipid (14:0-PC) to total protein ratio of the solution NMR samples was ~150/1 (mol/mol). Control homodimer samples with isotopically enriched wild-type EmrE only were prepared in a similar fashion. The sample buffer was 150 mM $Na_2HPO_4$ and 20 mM NaCl. The experiments with ethidium bromide and tetraphenylphosphonium were carried out at concentrations of 2.1 mM and 1.05 mM, respectively.

In order to assess whether a 1 hr incubation time at 37°C was sufficient to achieve complete mixing of the heterodimer, a control experiment was carried out by comparing NMR spectra after 1 hr and 19 hr of incubation at 37°C. The NMR spectra collected of these experiments showed that shorter and longer incubation times gave statistically the same ratios of monomer A and B peak intensities, which demonstrates that complete mixing was achieved with 1 hr (*Figure 3—figure supplement 1B*). Note that the 1 hr incubation time in DDM is consistent with the reduced dimer stability of EmrE in DDM detergent micelles compared to that in lipid bicelles or lipid bilayers (*Dutta et al., 2014a*).

## Solid-State NMR spectroscopy

Oriented sample solid-state NMR experiments were acquired using an Agilent DD2 spectrometer at a $^1H$ frequency of 600 MHz equipped with a $^1H/^{15}N$ double resonance probe manufactured by Revolution NMR (design of Peter Gor'kov; *Gor'kov et al., 2007*). Experiments were carried out using magnetically aligned lipid bicelle samples that were flipped with the addition of 3 mM $YbCl_3$ to orient the bicelle normal parallel to the magnetic field. PISEMA (*Wu et al., 1994*) spectra were acquired with SPINAL-64 $^1H$ decoupling during acquisition and phase modulated Lee-Goldberg (*Vinogradov et al., 1999*) (PMLG) $^1H$-$^1H$ decoupling in the indirect dimension. The radiofrequency field for SPINAL decoupling was 50 kHz, while the effective field for PMLG was 41.7 kHz. Spectra were acquired with ~1500 scans and 14 increments in the indirect dimension with a recycle delay of 3 s. The indirect dimension axis was corrected with the scaling factor of 0.82. The $^{15}N$ direct dimension was referenced to $^{15}NH_4Cl$ (solid) at 41.5 ppm.

## Solution NMR spectroscopy

Solution NMR experiments were acquired using a Bruker Avance III spectrometer at a $^1H$ frequency of 600 MHz equipped with a TCI cryogenic probe. $^1H/^{13}C$ HMQC experiments were acquired at 25°C using $^1H$ and $^{13}C$ spectral widths of 10,000 Hz and 4,000 Hz, respectively. The total acquisition ($^1H$) and evolution times ($^{13}C$) corresponding to $t_2$ and $t_1$ were 59.9 msec and 18.5 msec, respectively. Spectra were acquired with 12 or 24 scans with a recycle delay of 1 s.

## Quantification of population from heterodimer samples

The equilibrium shown in *Equation 1* corresponds to a heterodimer composed of wild-type EmrE and mutant:

$$WT_A \cdot mutant_B \; \rightleftharpoons \; mutant_A \cdot WT_B \tag{1}$$

Subscripts 'A' and 'B' correspond to monomers A and B in the asymmetric dimer. A simple calculation of the peak intensity ratio for A and B resonances in the heterodimer samples does not correspond to the equilibrium constant since the peak intensities contain a statistical fraction of homodimers ($f_{homo}$) and heterodimers ($f_{het}$) based on the ratio of isotopically labeled and natural

abundance proteins present for mixing. Note that $f_{homo}$ and $f_{het}$ consider only homodimers or hetero-dimers that contain the isotopically enriched protein. The observed peak intensities for monomer A ($I_{A,obs}$) and B ($I_{B,obs}$) signals in a heterodimer sample are given by Eqns. 2 and 3:

$$I_{A,obs} = I_A \ (f_{homo} + f_{het}\,p_A) \tag{2}$$

$$I_{B,obs} = I_B \ (f_{homo} + f_{het}\,p_B) \tag{3}$$

*Equation 2* divided by *Equation 3* gives:

$$\frac{I_{A,obs}}{I_{B,obs}} = \frac{I_A}{I_B} \ \frac{(f_{homo} + f_{het}\,p_A)}{(f_{homo} + f_{het}\,p_B)} \tag{4}$$

$I_A/I_B$ is the ratio of 'intrinsic' intensities of monomer A and B peaks obtained from the homodimer spectrum. This is needed since A and B peaks in the homodimer spectrum are not exactly the same. $f_{homo}$ and $f_{het}$ were set to 1/7 and 6/7, respectively, by assuming statistical mixing of the molar ratio of isotopically labeled wild-type to natural abundance L51I mutant in solution NMR experiments (i.e., 1 part labeled to 3 parts unlabeled). $p_A$ and $p_B$ are populations of wild-type in monomer A or B confor-mations when in a heterodimer with a mutant (see *Equation 1*). The addition of these populations is given by *Equation 5* and their ratio is the equilibrium constant (*K*) in *Equation 6*:

$$p_A + p_B = 1 \tag{5}$$

$$K = \frac{p_B}{p_A} \tag{6}$$

Substitution of $p_B$ from *Equation 5* into *Equation 4* gives the following:

$$\frac{I_{A,obs}}{I_{B,obs}} = \frac{I_A}{I_B} \ \frac{(f_{homo} + f_{het}\,p_A)}{(f_{homo} + f_{het}\,(1 - p_A))} \tag{7}$$

$p_A$ was solved from *Equation 7* for drug-free (pH = 9.1) and tetraphenylphosphonium bound forms of wild-type/L51I heterodimers using solution NMR HMQC spectra of wild-type EmrE and iso-topically labeled wild-type mixed with natural abundance L51I. The average and error range (derived from the standard deviation) of the populations were determined from the following residues that displayed well-resolved signals in each spectrum: Ile11, Ile54, Ile58, Ile62, Ile68, Ile88, and Ile101.

## Resistance assays on solid media

Growth inhibition assays were performed in pET Duet-1 vectors with constructs designed similar to that of *Rapp et al. (2007)*. All of the constructs are shown in *Figure 4—figure supplement 1*. EmrE$^{in}$ consists of three mutations relative to wild-type EmrE (R29G, R82G, S107K) and induces the N- and C-termini to face the cytoplasm. EmrE$^{out}$ consists of three mutations from wild-type (T28R, L85R, R106A) and induces the N- and C-termini to face the periplasm. EmrE$^{in}$ and EmrE$^{out}$ were placed in the second cloning site. An additional mutation to EmrE$^{in}$ and EmrE$^{out}$ constructs were made by mutating E14 to Q14. These constructs are also single topology and are referred to as E14Q$^{in}$ and E14Q$^{out}$. The L51I and I62L constructs are single site mutants of wild-type EmrE (UniProt P23895).

Luria-Bertani (LB) media and Luria agar powder for resistance assay were purchased from Research Products International. Each construct in the pET Duet-1 vector was transformed into BL21 (DE3) and grown at 37°C up to an OD$_{600}$ of ~1.5. The cultures were diluted to an OD$_{600}$ of 1.0 with fresh LB supplemented with carbenicillin (100 µg/mL) and were then serially diluted by 10-fold to achieve final dilutions of $10^0$ to $10^6$. All of the dilutions used fresh LB media containing 100 µg/mL carbenicillin. 3 µl of each dilution were pipetted onto plates containing 20 µM IPTG, 100 µg/mL car-benicillin, and 94 µg/ml ethidium bromide. Control experiments were carried out by plating cells onto LB agar plates containing 20 µM IPTG and 100 µg/mL carbenicillin. All resistance assays on solid medium were repeated at least two times.

## Resistance assays in liquid media

Liquid assays were performed using the same constructs as for the solid media resistance assay. *E. coli* BL21 (DE3) were transformed with the pET Duet-1 vectors and grown at 37°C until the $OD_{600}$ was ~1.5. The cultures were then diluted to an $OD_{600}$ of 1.0 with fresh LB supplemented with carbenicillin (100 µg/mL) and ethidium bromide (240 µg/mL). Cultures were incubated for 11.5 hr and the $OD_{600}$ was measured every hr up to 6 hr and one additional time 11.5 hr after the initial exposure to ethidium bromide. A full set of resistance assays in liquid medium were repeated three times. The error bars reflect the standard deviation for replicate trials.

## Ethidium bromide efflux assay

Mutant constructs used in resistance assay were transformed into BL21(DE3) and grown to mid log phase at 37°C ($OD_{600}$ ~1.0). The cells were spun down and diluted to the $OD_{600}$ of 0.1 in minimal media A (40 mM $K_2HPO_4$, 22 mM $KH_2PO_4$, 2 mM sodium citrate, 0.8 mM $MgSO_4$, and 7.6 mM $(NH_4)_2SO_4$, pH 7.0). The resuspended cells were treated with 80 µM carbonyl cyanide *m*-chlorophenyl hydrazine (CCCP) for 5 min. Ethidium bromide (10 µg/mL) was added to the cells and incubated for 30 min at 37°C while shaking. Cells were spun down for 10 min and the pellet was kept on ice until the florescence experiment. For the efflux assay, cells were resuspended in minimal media A with 10 µg/mL ethidium bromide. To initiate the assay, 0.2% (w/v) glucose was added. Control experiments were carried out by treating the cells in the same manner except without adding glucose. The fluorescence decays were measured with a Molecular Devices FlexStation 3 instrument using an excitation wavelength of 530 nm and an emission wavelength of 600 nm. The fluorescence was recorded over a time period of 3600 s. Each experiment was acquired in duplicate and repeated at least two times. The error bars reflect the duplicate experiments carried out on the same day.

## Acknowledgements

This work was supported by NIH (R01 AI108889) and NSF awards (MCB 1506420) to NJT. ML was supported from a Dean's Dissertation Fellowship from New York University. The NMR data collected with a cryoprobe at NYU were supported by an NIH S10 grant (OD016343).

## Additional information

### Funding

| Funder | Grant reference number | Author |
| --- | --- | --- |
| National Institutes of Health | R01AI108889 | Nathaniel J Traaseth |
| National Science Foundation | MCB1506420 | Nathaniel J Traaseth |
| National Institutes of Health | S10OD016343 | Nathaniel J Traaseth |

The funders had no role in study design, data collection and interpretation, or the decision to submit the work for publication.

### Author contributions

Maureen Leninger, Conceptualization, Data curation, Formal analysis, Writing—original draft, Writing—review and editing; Ampon Sae Her, Conceptualization, Data curation, Formal analysis, Writing—review and editing; Nathaniel J Traaseth, Conceptualization, Formal analysis, Writing—original draft, Writing—review and editing

### Author ORCIDs

Nathaniel J Traaseth https://orcid.org/0000-0002-1185-6088

### Decision letter and Author response

Decision letter https://doi.org/10.7554/eLife.48909.018
Author response https://doi.org/10.7554/eLife.48909.019

## Additional files

### Supplementary files
• Transparent reporting form
DOI: https://doi.org/10.7554/eLife.48909.016

### Data availability
All data generated or analyzed during this study are included in the manuscript and supporting files.

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
