## [Decision Letter]

Thank you for submitting your article "Inducing conformational preference of the membrane protein transporter EmrE through conservative mutations" for consideration by *eLife*. Your article has been reviewed by three peer reviewers, and the evaluation has been overseen by Olga Boudker as the Reviewing Editor and Senior Editor. The following individual involved in review of your submission has agreed to reveal their identity: Timothy A Cross (Reviewer #1).

In the manuscript, the authors examine the conformational equilibrium of homodimeric EmrE transporter. Individual EmrE promoters insert and assemble in anti-parallel manner and each assumes conformation A or confirmation B. Their structural transition from A to B and from B to A conformations leads to the transition between states open to the periplasm or cytoplasm. Because protomers are identical, there is no conformational bias toward either outward or inward facing states. The authors introduce single subtle mutations into a protomer and show that when such protomers assemble with WT protomers they demonstrate conformational preference for conformation A, while WT protomers assume preferentially conformation B. Now, when mutants are assembled with WT protomers whose insertion topology is fixed by additional mutations relying on "positive inside" principle, the authors presumably generate in cells assemblies that have preferential outward- or inward-facing configurations. They show that conformational preference for the inward-facing states results in active assemblies, while conformational preference for the outward-facing state leads to reduced activity. This is an elegant study that probes the role of the so-called "energy landscape" in defining the rate of transport. The reviews have found both the solid-state and the solution NMR spectroscopies to be well done and cell assays appropriate and informative. There were several aspects of the work that, however, raised concerns.

Essential revisions:

1) Key experiments in this manuscript cover ground that was already covered in the authors' 2016 Nature Chemical Biology paper (Gayen et al., 2016). It is a really interesting observation that a single mutation in only one subunit changes the energy landscape. This observation, originally reported in Gayen et al., is replicated using a different NMR strategy here. It would be critical for the authors to articulate what is different and new in the current study. It was also puzzling why different mutants were used in this paper compared to the 2016 paper. On the related note, the Introduction section was perceived as insufficient to introduce non-specialists to the EmrE class of transporters and to explain what the goals of the current experiments are.

2) The reviewers were surprised by how facile the preparation of the heterodimers was. Mixing for 1 hour at 37 °C is often insufficient to achieve subunit exchange in membrane proteins. Slower kinetics is typical for membrane proteins in general (Jefferson…Bowie, JACS, 2013), and has been reported for EmrE specifically (Rotem…Schuldiner JBC 2001). The authors should present evidence that complete mixing has been achieved. Further, fast kinetics also raises question whether monomers are present in the mix at significant levels during data collection. The authors should address this concern. Finally, there exists the possibility that the subunits actually exchange during the experiment. The experiments with labeled Tyr have very close to equal amounts of labeled and unlabeled protein, so the effect of subunit exchange would be expected to be pretty substantial. How would this change the results?

3) The reviewers further felt that the paper would be significantly strengthened if the authors took more quantitative approach to the analysis of populations. "Conformational preference" is an imprecise term. The authors should consider analyzing populations quantitatively to estimate free energies associated with preferred conformations. This is particularly pertinent because mutations are, indeed, very subtle and putting numbers on their effects would be informative.

4) The reviewers were surprised further by how detrimental to the bacterial survival the mutations were. How significant should be the population shift to lead to a complete loss of resistance to ethidium bromide? Even in the authors' simulations, the difference in uptake rate is only three-fold and it is difficult to reconcile such relatively subtle kinetic effects with complete loss of resistance. It would be important for the authors to comment on this more explicitly.

5) The Figure 5D is confusing. Why is there a conformational preference for the inward-facing state in EmrE^in^/EmrE^out^ dimer? Should not it, like WT, have no conformational preference? What is exactly depicted in the middle panel? Here, the preference should be for the outward-facing state shouldn't it? Perhaps, the figure should be remade to make the point the authors are making clearer. It might be better to show the entire transport cycle and point out which rates are likely affected by the asymmetry generated by mutations within the assemblies with EmrE^in^ or EmrE^out^ variants. Such depiction may also allow to clarify which conclusions are fully supported by the data and which are more speculative in nature.

Reviewer #1:

I found this work to be very clever and the results to be exciting. Both the solid-state and solution NMR spectroscopies were well done as were the assay studies. However, as a membrane protein spectroscopist and as someone who has not kept up with the EmrE literature, I found the start of the Results and Discussion text very difficult to comprehend. The Introduction did not include an introduction to EmrE, but only to the class of transporters – as a result I found the Results and Discussion section difficult to comprehend – a paragraph in the Introduction about EmrE in/out, EmrE A and B etc. would be helpful for a broader audience.

The study involves the functional characterization of two conservative mutants of an antiparallel dimeric transporter, EmrE. These mutants L51I and I62L result in preferential roles when paired with a WT monomer. Furthermore, these WT/mutant hybrids result in modified functionality. Clever mutation of EmrE by changing the inward facing charge on the structure results in an inward facing structure EmrE^in^ and an outward facing structure EmrE^out^, these mutants in combination displayed resistance to ethidium. Furthermore, the EmrE dimers of L51I and dimers of I62L display resistance to ethidium, but L51I with EmrE^in^ and I62L with EmrE^in^ both displayed reduced resistance to ethidium. However, the mutants paired with EmrE^out^ displayed WT resistance to ethidium. Based on the structural studies with NMR spectroscopy the authors were able to note that the conservative EmrE mutants have a preference for monomer A in the heterodimers with WT. For EmrE^out^ with a mutant the heterodimer preference is inward-open/cytoplasmic-facing conformation and for EmrE^in^ the preference was for the outward-open/periplasmic-facing conformation under the same drug free conditions. This led the authors to justifiably make two conclusions: 1) that the "outward-open/periplasmic (facing) state has a deleterious impact on the overall transport cycle" and 2) that the "inward-open/cytoplasmic facing conformation" gives no such deleterious effect. This is really an elegant detective story involving these subtle mutations at the ends of two TM helices that anchor an extramembrane loop.

Reviewer #2:

This manuscript from the Traaseth lab investigates the effect of a pair of conservative mutations, L52I and I62L, on protein conformation and drug export function in EmrE.

The best experiments in this manuscript cover ground that was already covered in the authors' 2016 Nature Chemical Biology paper (Gayen et al., 2016). It is a really interesting observation that a single mutation in only one subunit changes the energy landscape. This observation, originally reported in Gayen et al., is replicated using a different NMR strategy here. However, my overall impression of the manuscript is that the follow-up experiments are too observational and qualitative. There are a lot of aspects of this study that I find surprising (beyond the fact that a one methyl group alteration does anything at all!), and presenting these surprising observations without deeper digging is somewhat unsatisfying.

I'm surprised that the subunit mixing is so extensive. The 1 hr, 37 °C incubation that the authors use to achieve heterodimers suggests faster subunit exchange kinetics than is typical for membrane proteins in general (Jefferson…Bowie, JACS, 2013), or what has been reported for EmrE specifically (Rotem…Schuldiner JBC 2001). Are there monomers in equilibrium too? I would feel more comfortable if the authors had some way to assess monomer vs. heterodimer vs. homodimer populations in their samples. Moreover, I do not know what temperature/acquisition time was used to collect the NMR data shown here (or how the kinetics differ in detergent vs. bicelles vs. solid state bilayer), but if subunit mixing reaches equilibrium in an hour, there certainly exists the possibility that the subunits actually exchange during the experiment. The experiments with labeled Tyr have very close to equal amounts of labeled and unlabeled protein, so the effect of subunit exchange would be expected to be pretty substantial. How would this change the results?

The term "conformational preference" is used throughout. Is there some reason the authors don't frame this as conformational equilibrium? I assume that this is some kind of equilibrium process - the heterodimers also go through conformational exchange, except that now the equilibrium more strongly favors the mutant subunit in one state, and the wildtype subunit in the other. This is one area where a more rigorous analysis of equilibrium constants would be very useful. Can the differences in peak volume be measured/reported? A numerical ratio comparing two residues in the "A" and "B" positions would be a better metric than trying to evaluate how peak intensity and width change from the bird's eye view. For example in Figure 3A, I'm just not able to evaluate the author's assertion that the heterodimer has shifted the population towards "B." There aren't, to my eye, enough assigned peaks and direct A:B comparisons.

I'm also very surprised that the EmrE-in/mutant heterodimers are so sensitive to ethidium (as sensitive as abolishing one of the glutamates!). Survival changes by four orders of magnitude! While this paper is short on quantitation, in the previous Nat. Chem Biol paper, the authors estimated that the heterodimer favors one conformation over the other by about 2:1 (compared to 1:1 for WT). In terms of Delta G, this is a small change. The authors agree that their data does not indicate that the mutant locks the transporter in one configuration, and have shown that conformational exchange still occurs for the homodimeric mutant. So I'm puzzled by the extent to which this mutant disables drug export.

In addition, I have some concerns about data presentation. Even though the authors are forthright about doing so, I don't like the maneuver of cutting the contour levels to obscure non-A or non-B peaks in Figure 2. I think the spectra included in the supplement, which show both populations of peaks, are a more faithful visual representation of the actual data. I also don't like showing part of the Figure 1B spectrum at 4x noise, and the rest of the spectrum at 5x noise. I'm guessing that some unexplainable peaks appear elsewhere in the spectrum at 4x noise, which could indicate sample heterogeneity.

Reviewer #3:

In the manuscript, the authors examine the conformational equilibrium of homodimeric EmrE transporter. Individual EmrE promoters insert and assemble in anti-parallel manner and each assumes conformation A or confirmation B. Their structural transition from A to B and from B to A conformations leads to the transition between states open to the periplasm or cytoplasm. Because protomers are identical, there is no conformational bias toward either outward or inward facing states. The authors introduce single subtle mutations into a protomer and show that when such protomers assemble with WT protomers they demonstrate conformational preference for conformation A, while WT protomers assume preferentially conformation B. Now, when mutants are assembled with WT protomers whose insertion topology is fixed by additional mutations relying on "positive inside" principle, the authors presumably generate in cells assemblies that have preferential outward- or inward-facing configurations. They show that conformational preference for the inward-facing states results in active assemblies, while conformational preference for the outward-facing state leads to reduced activity. This is an elegant study that probes at the role of the so-called "energy landscape" in defining the rate of transport. I found it overall compelling, but confusing in interpretation and presentation.

In particular, I found Figure 5D and corresponding discussion confusing. Why is there a conformational preference for the inward-facing state in EmrE^in^/EmrE^out^ dimer. Should not it, like WT, have no conformational preference? What is exactly depicted in the middle panel. Here, the preference should be for the outward-facing state, right? I think that this figure should be remade. It might be better to show the entire transport cycle and point out which rates are likely affected by the asymmetry generated by mutations within the assemblies with EmrE^in^ or EmrE^out^ variants. My first take on this is as follows: the conformational exchange of the drug-bound transporter is comparatively fast and is not rate-limiting to the cycle. Thus, it is the rate of the transition of the substrate-free transporter that determines the overall rate. In the transport cycle, where drug extrusion is coupled to influx of protons, the outward to inward translocation of the protonated transporter would determine the overall rate. Thus, if mutation favors the inward-facing conformation of Mut/EmrE^out^ dimer, it might do so through accelerating the rate of outward to inward transition and therefore speeding up the cycle. Or, alternatively, it could slow the reverse transition from outward to inward and increase the amount of time for the drug to be successfully released into the periplasm.

---

## [Author Response]

Essential revisions:1) Key experiments in this manuscript cover ground that was already covered in the authors' 2016 Nature Chemical Biology paper (Gayen et al., 2016). It is a really interesting observation that a single mutation in only one subunit changes the energy landscape. This observation, originally reported in Gayen et al., is replicated using a different NMR strategy here. It would be critical for the authors to articulate what is different and new in the current study. It was also puzzling why different mutants were used in this paper compared to the 2016 paper. On the related note, the Introduction section was perceived as insufficient to introduce non-specialists to the EmrE class of transporters and to explain what the goals of the current experiments are.

The novelty of the current work is to show how the energy landscape is perturbed by a single mutation for several states in the transport cycle and to provide functional support for these in vitro observations. The previous publication in 2016 reported a change in the conformational equilibrium for only the tetraphenylphosphonium bound state and did not report any functional significance for the observation. The revised submission underscores the structure-function relationship we aim to develop while clearly delineating what was previously studied and what is novel about our current work (see Introduction, second and third paragraphs).

With regard to mutations, our initial report in 2016 utilized the I54L and I62L mutations. Since this publication, we discovered that the L51I mutation has the most prominent effect in influencing the conformational equilibrium when paired with a wild-type monomer. For this reason, the majority of our studies were performed with this mutation. The revised version states this point in the Introduction section (second and third paragraphs).

As referenced above, the Introduction has been revamped to provide additional background information on EmrE and to more clearly articulate the goals of our experiments. We also highlight what was accomplished in the 2016 paper, which will enable the reader to identify the novelty of our current work.

2) The reviewers were surprised by how facile the preparation of the heterodimers was. Mixing for 1 hour at 37 °C is often insufficient to achieve subunit exchange in membrane proteins. Slower kinetics is typical for membrane proteins in general (Jefferson…Bowie, JACS, 2013), and has been reported for EmrE specifically (Rotem…Schuldiner JBC 2001). The authors should present evidence that complete mixing has been achieved. Further, fast kinetics also raises question whether monomers are present in the mix at significant levels during data collection. The authors should address this concern. Finally, there exists the possibility that the subunits actually exchange during the experiment. The experiments with labeled Tyr have very close to equal amounts of labeled and unlabeled protein, so the effect of subunit exchange would be expected to be pretty substantial. How would this change the results?

The EmrE dimer has been shown to be at least 1000-fold more stable in lipids compared to dodecyl maltoside (DDM) detergent micelles (Dutta et al., 2014). This is pertinent to the monomer exchange experiments reported by Rotem et al., since they were carried out by solubilizing *E. coli* membranes with DDM (i.e., native lipids were present). This is in contrast to the preparation of our heterodimer samples, which were carried out on purified proteins in DDM (i.e., few native lipids present). In our hands, purified EmrE in DDM in the drug-free state cannot be heated above ~40 °C without noticeable precipitation after 1 hr. This likely reflects the presence of monomers during the mixture procedure that would increase kinetics relative to the Rotem et al. study. To provide evidence that the mixing is complete, we prepared two parallel samples, where one was incubated at 37 °C for 1 hr and the other was incubated at 37 °C for 19 hrs. Solution NMR samples were prepared in the same way. The ratio of subunit A and B signals was statistically the same between the samples, which supports that complete mixing has been achieved (see Figure 3—figure supplement 1).

The next question was whether there are monomers in our NMR samples. The EmrE dimer is at least 1000-fold more stable in lipid bilayers and ~100-fold more stable in lipid bicelles compared to DDM detergent micelles (Dutta et al., 2014). Furthermore, NMR spectra show only two peaks per residue, where each set of signals stems from one monomer in the asymmetric dimer. This has been confirmed by NMR spectroscopy on cross-linked dimers that give the same peak position as wild-type samples (Cho et al., 2014) and also agrees with the asymmetry seen in cryoEM (Tate et al., EMBO J, 2001, 20, 77-81) and X-ray datasets (Chen et al., 2007). This fact is further underscored by the heterodimer experiments presented in the current work, where each monomer in the heterodimer gives rise to one set of signals.

The last question is whether exchange of labeled and unlabeled proteins occur during the NMR experiments. While exchange is theoretically possible, to the extent that it occurs, it would not change any of our conclusions for two reasons. First, exchange in bicelles would be a slow process on the NMR timescale since the peak positions do not move relative to those of the respective homodimers. This is consistent with the stability of the EmrE dimer reported in bicelles (Dutta et al., 2014). Second, the system is at equilibrium as established from the 1 hr and 19 hr mixing experiments (Figure 3—figure supplement 1), so any subunit exchange would be offset to maintain identical peak intensities of monomers A and B. The fact that the Tyr labeled PISEMA experiments used a lower ratio of isotopically enriched protein to natural abundance protein only changes the expected statistical fraction of homodimers and heterodimers in the sample.

3) The reviewers further felt that the paper would be significantly strengthened if the authors took more quantitative approach to the analysis of populations. "Conformational preference" is an imprecise term. The authors should consider analyzing populations quantitatively to estimate free energies associated with preferred conformations. This is particularly pertinent because mutations are, indeed, very subtle and putting numbers on their effects would be informative.

In the resubmission, we quantified the relative populations in the heterodimer equilibrium corresponding to wild-type and the L51I mutant:

WTA∙mutantB⇌mutantA∙WTB

The calculation is described in detail in the Materials and methods section “Quantification of Populations from Heterodimer Samples”. The populations are reported in the Results section “Assessing Conformational Preference for Different States within the Transport Cycle” for the drug-free deprotonated and the tetraphenylphosphonium bound heterodimers derived from solution NMR experiments shown in Figure 3. The populations of wild-type assuming monomer B in the heterodimer were estimated to be 96% for the apo form and 86% for the tetraphenylphosphonium bound form. These values correspond to free energies of ~1.8 kcal/mol for the apo form and ~1.1 kcal/mol for the tetraphenylphosphonium-bound form. In the Results section, we also report the error range which reflects the deviation among residues used for the calculation.

4) The reviewers were surprised further by how detrimental to the bacterial survival the mutations were. How significant should be the population shift to lead to a complete loss of resistance to ethidium bromide? Even in the authors' simulations, the difference in uptake rate is only three-fold and it is difficult to reconcile such relatively subtle kinetic effects with complete loss of resistance. It would be important for the authors to comment on this more explicitly.

We realized that the drug binding constants for the heterodimer in the simulation is not the same for wild-type(A)/L51(B) and L51I(A)/wild-type(B). In the resubmission, the binding constants were adjusted by the ratio of the equilibrium constants governing the AB to BA transition for the heterodimer in the absence and presence of the drug. The revised simulations (Figure 5—figure supplement 2) show that the concentration of drug into the liposome mutant/EmrE^out^ is slightly greater than for the mutant/EmrE^in^ sample. However, there is a notable difference in the initial rate of transport. Based on these simulations, we hypothesized that the difference in the transport rate by mutant/EmrE^in^ and mutant/EmrE^out^ would lead to a greater cytoplasmic accumulation of ethidium for mutant/EmrE^in^. The higher accumulation would display a reduced minimum inhibitory concentration, causing mutant/EmrE^in^ to be more susceptible at the ethidium concentration used in growth inhibition assays. To provide evidence that the overall rate of ethidium transport was different between mutant/EmrE^in^ and mutant/EmrE^out^ in vivo, we carried out an ethidium efflux assay by following the intrinsic fluorescence of ethidium. In this assay, the membrane potential is disrupted with the ionophore carbonyl cyanide mchlorophenylhydrazone (CCCP), causing the cytoplasmic ethidium concentration in the cell to increase and give rise to higher fluorescence. Upon addition of glucose and removal of CCCP, the membrane potential is re-established and the fluorescence decreases as ethidium is effluxed out of the cytoplasm.

The ethidium efflux assay was carried out for L51I/EmrE^out^, L51I/EmrE^in^, and a control plasmid not expressing EmrE. These results showed that the rate of ethidium efflux was ~3-fold faster for L51I/EmrE^out^ than for L51I/EmrE^in^ (Figure 5D), which is in agreement with the simulation results. Furthermore, the fluorescence value at which ethidium fluorescence levels off was lower for L51I co-expressed with EmrE^out^, indicating a reduction in ethidium accumulation relative to L51I co-expressed with EmrE^in^ (Figure 5E). Overall, these new results support the hypothesis that the L51I/EmrE^out^ heterodimer is able to achieve a lower cytoplasmic ethidium concentration, which enables it to grow at higher ethidium concentrations relative to L51I/EmrE^in^. Based on these data, we conclude that the altered kinetic rate constants between L51I/EmrE^in^ and L51I/EmrE^out^ give rise to the ability of the latter to confer resistance at higher ethidium concentrations. Note that previous work by Paixão et al. (Paixão et al., 2009) found that an ethidium efflux rate difference of 1.6-fold between wild-type *E. coli* and its efflux pump knockout strain led to a 30-fold difference in the minimum inhibitory concentration. Therefore, there is literature precedence that a somewhat modest efflux rate difference can have a deleterious effect on bacterial growth. The underlying reason stems from the fact that ethidium accumulation in the cell ultimately leads to irreversible DNA binding that impacts bacterial growth (Jernaes et al., 1994; Lambert et al., 1984).

5) The Figure 5D is confusing. Why is there a conformational preference for the inward-facing state in EmrE^in^/EmrE^out^ dimer? Should not it, like WT, have no conformational preference? What is exactly depicted in the middle panel? Here, the preference should be for the outward-facing state shouldn't it? Perhaps, the figure should be remade to make the point the authors are making clearer. It might be better to show the entire transport cycle and point out which rates are likely affected by the asymmetry generated by mutations within the assemblies with EmrE^in^ or EmrE^out^ variants. Such depiction may also allow to clarify which conclusions are fully supported by the data and which are more speculative in nature.

Figure 5F (previously 5D) was revised to show a simple depiction that reflects the observed bias in vitro. Namely, EmrE^in^/EmrE^out^ has no skewed equilibrium, the mutant/EmrE^in^ favors the periplasmic-facing conformation, and the mutant/EmrE^out^ favors the cytoplasmic-facing conformation. We also included the EmrE transport cycle in Figure 5G proposed by Robinson et al., 2017, that was used for the simulations. Within this figure, we indicate steps influenced by the change in equilibrium constant measured in the heterodimer samples. Rate constants we predict to be most affected are indicated with double asterisks, while more minor changes are indicated with a single asterisk.